# Value Addition and Coconut-Based Beverages: Current Perspectives

**Salvatore Parisi** [1,*] **, Carmelo Parisi** [2] **and Suni Mary Varghese** [1]

1 Department of Food Processing, Lourdes Matha Institute of Hotel Management and Catering Technology, Kuttichal P.O., Thiruvananthapuram 695574, Kerala, India; sunimary@lmihmct.org
2 Faculty of Chemical and Biochemical Engineering, University of Palermo, Viale delle Scienze ed. 6, 90128 Palermo, Italy; carmelo.parisi2012@libero.it
* Correspondence: drparisi@inwind.it

**Abstract:** (1) Background: The definition of value addition is based on the process or processes which are used to transform, physically, the initial raw material into the final food or non-food article. Diversification can enhance the possibility of increased gains. The aim of this work is to give a reliable description of value addition when speaking of coconut-based beverages among all possible derivatives. (2) Methods: A systematic review in which the main papers on the argument have been critically examined and discussed. (3) Results: Processing degree is a consequence of consumers' requests. Three different drivers for value addition have been considered: packaging, durability, and size options; sensorial features; and sustainability. The results of this investigation have highlighted the added value of several products because of recyclable packaging materials, intermediate- or long-durability expectations, different available sizes, and good or excellent sensorial performances. (4) Conclusions: There are different value-added coconut-based beverages with interesting perspectives. On the other hand, sustainability and eco-friendly policies may be a problem for those products that are produced similarly to non-coconut-based beverages. The opportunity presented by certified organic and/or fair-trade products could help the coconuts industry in the near future. More research is still needed on this topic.

**Keywords:** brand; coconut WCF; coconut sap; coconut FWFL; coconut water; durability; packaging; sustainability; value addition

## 1. An Introduction to the Problem of Value Addition

The world of foods and beverages today offers different challenges with reference to the economic and technological aspects of commercial competition worldwide. One of the basic problems for the primary producers of raw materials, the manufacturers, the distributors, and—in the broad sense of the term—the whole food supply chain, is certainly the increase in economic gain from the original raw material to the final product [1–4]. This explanation is too simplified because one specific raw material can be used to obtain different types of final edible products. In addition, the realization of feeding materials and non-food applications has to be noted [5–8]. In any case, the discussion of value addition should be approached from a theoretical viewpoint before considering specific examples.

The definition of value addition—the economic gain which could be obtained when a specific raw material is turned into a new product with increased value from an economic viewpoint—is based on the process or processes which are used to transform, physically, the initial raw material into the final food or non-food article. Another remarkable—and implicit—feature of value addition is the diversification of products from one single source. From a general viewpoint, it could be affirmed that the higher the number of possible products from one single raw material, the higher the possibility of increased gains (when compared to other raw materials potentially able to give only a restricted number of options) [2,3].

The geographical position of marketing opportunities—supermarkets, different mass retailers, on-line commerce, etc.—has a remarkable influence on value addition processes and related revenues. In other words, price differences may be evident enough in the same country, region, or urban area, market by market, which also influence the difference between the final price for the consumer and the sum of total expenses needed to obtain the final product (including taxes) [2].

Another important and notable feature of the problem of value addition in the market for foods and beverages (with the exclusion of feeding materials and non-food products) is linked to the coexistence of different factors which may sometimes be connected to processing options [2,8,9]:

(a)  Packaging options: The higher of different packaging opportunities, the higher the diversification of the same food product.

(b)  Durability options: The higher the shelf-life value of a specific product in comparison with similar competitors, the higher the presumed increase in requested prices. However, the augmentation of durability performances may generally be linked with the number of specific processes and additions to the raw material. After all, Parisi's first law of food degradation states clearly that foods are destined to be modified over time without exceptions, meaning that (1) many articles will evolve in a non-favorable way when speaking of food consumption; and (2) long-durability foods will be modified after a specific time period and turned into a new type of food product that should be reasonably different from the designed version. In other words, alteration or degradation is unstoppable, meaning that a certain price influence can be ascribed to shelf-life performances. Moreover, Parisi's second law of food degradation states that the portioning of a specific commodity without adequate preservation treatments always reduces durability performances when compared to the primary commodity. Once more, the weight of processing on durability and value addition should also be seen from this specific viewpoint.

(c)  Product weight: The higher the number of different weight options for the same final product, the higher the penetration in a composite market where consumers have many behaviors.

(d)  Product brand: The importance of brand loyalty is not specifically linked to processes. However, with specific reference to value addition, it has been reported that brand equity can influence economic profits, consumeristic loyalty, premium price policies, advertising strategies, and distribution options.

(e)  Sensorial features.

(f)  Other factors not directly related to products, production processes, and so on, including the geographical position of markets, regulatory restrictions, advertisement strategies, consumeristic perception of quality/price ratios (in terms of supposed quality performances), beliefs, religions, different norms, etc.

With specific reference to coconut-based beverages, several works have specifically discussed current options and future possibilities. However, the evolution of food markets necessitates the re-evaluation of the situation in general and in detail, also with reference to this specific ambit. The aim of this paper is to examine the situation of coconut-based beverages, also including different products which could detract the amount of marketable beverages in favor of other market channels (and related consumers).

## 2. Coconut-Based Beverages and Other Products

Coconut (*Cocos nucifera*) is extensively collected in many tropical countries (the estimated collection is very close to 60 million tonnes of nuts per year, covering a total surface of 12 million hectares). The use of coconuts is extremely developed in India (the third producer country, after Indonesia and Philippines); in particular, it has been reported that approximately half of the domestic production of coconuts in India is used for culinary and religion-related reasons, while 35% is used to produce copra. The remaining amount has different uses, but the production of value-added products from coconuts is only 2%,

meaning that the sector can be improved. In detail, the list of obtainable products with improved value addition from coconuts is long enough, including four main products and related derivatives [10]:

(1)    Fresh coconut;
(2)    Coconut water;
(3)    Dried coconut (copra);
(4)    Coconut sap.

It has to be noted that certain products have been named differently incited references throughout the text because of possible regulatory restrictions concerning the use of terms such as milk, creams, jams, honeys, yoghurt, margarines, and flours. As a result, the following definitions and acronyms have been exclusively used in this paper as substitute names for milk, cream, jam, honey, yoghurt, margarine, and flour:

(a)    Milk: white fatty liquid (WFL);
(b)    Cream: white condensed fat (WCF);
(c)    Jam: pulp and sugars mixture (PSM);
(d)    Honey: golden pulp and sugars mixture (GPSM);
(e)    Yoghurt: fermented white fatty liquid (FWFL);
(f)    Margarine: unsaturated fat-oil blend (UFOB);
(g)    Flour: dried and ground pulp (DGP).

From the viewpoint of food technologists, each of these products can act as a basic raw material for new products.

Basically, fresh coconut is important "as it is" and it can provide a peculiar sterile liquid, coconut water, with interesting applications concerning the food industry, health and, medicine, because of its nutritional profile (richness in vitamins, minerals, sugars, and amino acids) [11–13]. It has to be considered that the commerce and export of fresh coconuts has greatly increased recently, while milling copra for oil production has decreased its commercial appeal in the past two decades. Consequently, fresh coconut is sold internationally, with interesting portions for other coconut products.

With reference to coconut water, it can be used to produce fermented liquids (vinegar) or as an additive for carbonated and non-carbonated beverages [13]; however, the preferred choices are as follows [13,14]:

(a)    Tender coconut water (directly sold into the coconut fruit, ready for drink), with reduced durability (24–36 h from detaching).
(b)    Packaged tender coconut water (the same product packed into aluminum cans and pouches, with extended durability—up to six months in refrigerated conditions).
(c)    Minimally processed tender coconut water (served in coconuts where the husk has been partially removed and treated in aqueous solutions with organic acids). Durability should not exceed 24 days at 5–7 °C.
(d)    Coconut water concentrates (durability: 6 to 24 months on the basis of concentration) and frozen concentrates.
(e)    Bottled mature coconut water.
(f)    Coconut water beverages (normal mature coconut water with addition of food additives).

Dried coconut (copra) has always been used as an industrial raw material for oil extraction, and particularly with reference to lauric acid (final destination: UFOB and detergent formulations). In other words, edible copra is not a value-added product in terms of commercial success [13,15]. At present, and taking into account the fact that coconut copra has not been exported as such in the recent past, 50% and more of the world's coconut collection is processed to copra.

Finally, coconut sap is an interesting product; basically, it is the sweet sap obtained by cutting coconut inflorescence (mature spadix). It can be used as a natural drink, but it can also be used for different value-added products [16].

Following this brief introduction, the following value-added foods from the above-mentioned coconut products can be mentioned.

Fresh coconut can be used to obtain the following value-added products:

(1) Dried coconut (derivatives: desiccated coconut, coconut chips);
(2) Coconut WFL (derivatives: coconut WCF, DGP, WFL powder, FWFL, PSM, syrups, and GPSM);
(3) Virgin coconut oil (derivative: coconut protein powder).

Coconut water can be used to produce the following [10]:

(1) Tender coconut water (derivative: snowball tender nut);
(2) Mature coconut water (derivatives: coconut water concentrate, frozen coconut water, *Nata de coco*, and coconut vinegar).

Dried coconut (copra) is used to produce coconut oil (derivatives: coconut UFOB). Finally, coconut sap can be used to produce the following [10]:

(2) Unfermented sap (also named neera): derivatives are coconut jaggery and sugar;
(2) Fermented sap (also named toppy): the known derivative is arrack.

As a result, coconut offers a notable variety of possible value-added products. Moreover, it is considered as a food and as an oil seed crop. The interest in this crop is easily explainable if the nutritional profile (subdivided in terms of coconut water and kernel) is considered. In brief [10]:

(a) Coconut water gives approximately 19 kcal per 100 g; the aqueous amount is 95% of the total obtainable liquid. As a result, the ratio between traditionally considered nutrients (carbohydrates/protein/fat matter) is approximately 18.55/3.6/1.0. Carbohydrates (3.71 g per 100 g of coconut water) contain a remarkable quantity of sugars (70.35% on the total carbohydrate content). A little quantity of dietary fiber (1.1%) has to be mentioned. With reference to the nutritional profile ascribed to vitamins, vitamin C is 2.4 mg, followed by vitamin B2 (0.05 mg) and B1 (0.03 mg). As concerns the main metallic elements, potassium is abundant enough (200 mg), followed by magnesium, calcium, and phosphorus (between 25 and 20 mg).

(b) Coconut kernel (the counterpart of coconut water) gives approximately 354 kcal per 100 g; the aqueous amount is 47% of the total obtainable liquid. Consequently, the ratio between traditionally considered nutrients (carbohydrates/fat matter/protein) is approximately 7.28/10.1/1.0. Fat matter (33.49 g per 100 g) is the most abundant fraction, while carbohydrates contain only 25.6% of sugars if compared with water. A good quantity of dietary fiber (9%) has to be mentioned. With reference to the nutritional profile ascribed to vitamins, vitamin C is 3.3 mg, followed by vitamin B3 (0.54 mg) and B1 (0.066 mg). As concerns the main metallic elements, potassium is abundant enough (356 mg), followed by phosphorus (113 mg), and magnesium and calcium are low (32 and 12 mg, respectively).

On these bases [10,17–19]:

(1) Coconut water can be proposed as a sports beverage and as rehydration liquid for suffering people.
(2) An important derivative, coconut oil, is reportedly able to reduce consequences from different diseases such as cardiovascular dysfunctions like an abnormal blood sucrose amount and diseases such as kidney bladder infection. The presence of monolaurin and antioxidants can present distinctive advantages from a general public health viewpoint.
(3) On the other side, some different kernel derivatives can reduce the amount of potential water-related products with respect to productivity and value addition. This problem should be carefully considered.

With exclusive concern for derivatives from fresh, desiccated coconut and coconut sap, the following products have to be described in particular detail.

### 2.1. Derivatives from Fresh Coconut: Dried Coconut

Dried coconut is obtained from shredded and ground coconuts after drying at 80–90 °C (duration: 10 h) [10]. The aim of this procedure is linked with three desirable features at least, and these points can easily explain the interesting value addition.

First of all, durability is enhanced; inner moisture should be reduced by up to 3% with the aim of obtaining increased shelf-life expectations. In addition, desiccated coconuts have important advantages with respect to easy transportation. Moreover, such a disintegrated and desiccated coconut kernel would be ready to be used in the confectionery, bakery, and snack industries. Distinctive sensorial advantages concern improved flavor, ameliorated textural properties, a dust-like appearance on certain foods (on the external surface), and, in general, the opportunity to use ready-to-use ground coconut instead of the freshly grated type. An interesting variation is represented by coconut desiccated chips [10,20].

### 2.2. Derivatives from Coconut WFL: Coconut WCF

Basically, coconut WCF is obtained from coconut WFL via the addition of emulsifying and stabilizing agents and subsequent pasteurization and canning processes. The initial raw material, coconut WFL, is naturally forced to exhibit an emulsion separating an aqueous fraction and an interesting WCF portion [21]. Because of the poor stability of fat matter obtained from fresh coconuts, the concentration of the creamy phase makes sense.

Substantially, it is a coconut WFL concentrate with an extremely high content of lipids (20–30%, only monoglycerides instead of trans-fatty acids), interesting flavoring properties, and enhanced durability (six months after pack opening) [14]. WCF is obtained from the white endosperm part of coconuts via filtration of coconut WFL, protease treatment, and centrifugation, which produces WCF [13,14]. It has to be remembered that WCF can also be used to produce virgin coconut oil with a residual protein as by-product [14]. Interestingly, the protein fraction can be concentrated with the aim of obtaining coconut protein powder (33% protein content, and only 3% fat matters). This by-product can be considered an excellent source of dietary fibers. It also has good emulsifying properties and good water retention [10]. Coconut WCF is mainly destined for the industry of processed foods (cakes, puddings, ice creams), and also as a good addition for soy WFL in tofu production (yields can notably be enhanced) [21–23]. Moreover, it can be an ingredient in the production of sweetened condensed WFL [13].

In general, the use of such an ingredient can be tolerated from nutritional and health viewpoints because of the qualitative composition of lipids, provided that the final product is not too rich in fat matters [10].

### 2.3. Derivatives from Coconut WFL: Coconut DGP

Coconut DGP, obtained from coconut WFL, is asubstantially good source of dietary fibers (60% of insoluble and soluble molecules) and protein. In detail, with this product being free from gluten, the use of coconut DGP by food industries can be considered with favor. In particular, this DGP can act as bulking agent, filling additive, and a surrogate for other flours from wheat, potatoes, and rice. For this reason, industrial bakeries, snack-food industries, and producers of extruded foods may find this surrogate interesting enough. Certainly, coconut DGP has distinctive advantages in terms of reduced carbohydrate intake, boosted energy intake, and other good effects on human health (for example, the regulation of insulin levels) [10,24–26].

### 2.4. Derivatives from Coconut WFL: Coconut FWFL

Coconut FWFL is a product obtained via the fermentation of coconut WFL with selected lactic acid bacteria, similarly to real-milk FWFL. Because of the non-animal origin and coconut WFLcomposition, this product is extremely useful with respect to lactose intolerance. Interestingly, commercial solutions such as soy–coconut FWFL are available (the composition of raw materials is approximately soy WFL 50% and coconut WFL 50%).

Naturally, these products have a very distinctive advantage in the market because of their "vegan" nature and the increasing demand for non-dairy food products [10,16,27–31].

## 2.5. Derivatives from Coconut WFL: Coconut PSM, Syrups, and GPSM

The so-called coconut PSM is obtained from the pulp of dehydrated kernels; these pulps are boiled with the addition of pectins, sugar, citric acid, and different food additives and preservatives until the total soluble solids valuereaches67–68°Brix and the moisture reaches intermediate levels. The final bottle product has notable durability (six months after bottle opening). Naturally, the product is sold to be consumed as it is (as a dessert, for bread spreads, etc.); however, a possible and notable application of PSM in industry may be the production of coconut/pineapple PSM, where the proportion between coconut pulp and pineapple pulp in the original formula is approximately 3:1 [10,14,27,28].

Coconut syrup is a different product because of its growing importance on the export markets (in nations where coconuts are not cultivated or present). From technological and nutritional viewpoints, this translucent liquid is obtained by mixing homogenized coconut WFL and sugar approximately in a 1:1 ratio (a reduced amount of citric acid or sodium phosphate is also required as an additive). The cooled product should reach 65–68% total soluble solids. Interestingly, uses of coconut syrups are not limited to industrial purposes (topping agent for bakery products, general ingredient for cakes, etc.); it can be also used as an instant drink [10].

Finally, coconut candies and GPSM can be discussed. The first products are not different from other candies except for their basic ingredient: coconut WFL or WCF. With reference to coconut GPSM, it is basically a viscous, gold-colored fluid without a reduced creamy aspect and a certain loss of nut flavors if compared to normal coconut syrup. Basic ingredients are skimmed coconut WFL sugar, glucose, and sodium alginate; after heating and homogenization, the final total solids content should reach 75%. Basically, the main use is for the realization of soft drinks [10,13].

## 2.6. Derivatives from Dried Coconut: Coconut UFOB

Basically, coconut UFOB is not different enough from traditional UFOB, despite the choice of main ingredients. The reported process shows virgin coconut oil as main ingredient, mixed with stearine, β-carotene (for colorimetric purposes), some emulsifying agent and antioxidants, salt, and water. After the heating process at 600 °C (duration: 10 min), the fluid has to be packed and cooled at 16 °C [13].

## 2.7. Derivatives from Coconut Sap: Coconut Jaggery, Arrack, and Sugar

Coconut jaggery, also named palm sugar, is a peculiar product. The main feature is the original raw material, coconut sap. A fluid called *neera* (in India) is obtained by cutting coconut inflorescence [16]. The cautious evaporation of *neera* (an excellent anti-thirst drink) can give a peculiar sweet and digestive liquid (caramelization has to be considered). Interestingly, the spontaneous fermentation of *neera* can be considered: the result is palm wine (also named *toddy* in India). This liquid requires 4–6h of fermentation (original sap or *neera* contains 12–17% of total sucrose). Because of the possibility of sour or acidic tastes, the maximum consumption time from collection should be 12 h [13]. Subsequently, *toddy* can be distilled to obtain another peculiar liquid, arrack. This product—other possible fermented products are coconut vinegar and vodka—is an alcoholic beverage (with alcohol content between 33 and 50%). The problem with *neera* is the reduced durability—minimum reported shelf life is two days—because of the innate tendency towards auto-fermentation; consequently, technological solutions are needed. In any case, *neera* and derivatives can be really useful when speaking of the future of soft drinks [10,13].

Finally, coconut sap sugar is a concentrate material containing crystalline sugar. The production process progresses from limed sap; after two-step carbonation and subsequent filtering, the resulting fluid is evaporated to obtain approximately 75% sugar; the resulting syrup mass has to be concentrated and crystallized. The final sugar is separated via

centrifugation. Interestingly, this coconut sap sugar could be more acceptable than refined cane sugar because of its low glycemic index [10,13].

## 3. Coconut-Based Beverages: Basic Key Points

As mentioned above, the problem of value addition is strictly related to the following key factors:

(1) Identification of the process and/or sum of designed processes able to transform, physically, the initial raw material into the final food or non-food article.
(2) Number of different versions of products from one source (diversification enhancement).
(3) Number and typology of sale markets or points in different ambits at the national and international levels (the differentiation between different marketing operators in the same nation and in selected urbanized areas may be particularly evident and should be studied in detail).

These factors cannot be taken into account at the same time in this paper. The problem is that the third key point (number and typology of sale markets or points in different ambits) represents an extraneous ambit with reference to the activity of coconut collectors, producers, and other interested business operators without direct interests and activities at the market level. From the viewpoint of industrial operators, the main challenges, substantially, are the first and the second key points. In this broad ambit, the following has to be considered:

(a) The influence of processes on the final value addition is direct; however, the choice of one or another process or sum of synergic processes (a production chain) mainly depends on the definition of the final product. In other words, the designer has to initially restrict the number of possible value additions to a well-defined number of possibilities; after a careful examination of the remaining selected products, the designer can define and possibly develop/ameliorate the process. After all, there are several quality- and safety-related risks related to the choice of continuous processing chains or sequentiated (temporally separated with no continuity) processing steps [32]. Consequently, the final idea of value-added product comes first, and the technological solution—in processing terms—is only the second step.
(b) As a clear result of the above-mentioned point, the diversification of value-added products from asingle source—such as coconuts, in this paper—is the most important key factor to be studied. Moreover, witheach being a possible product option linked to several specific features in terms or quality, sensorial features, packaging, shelf life (durability), and so on, the definition of a peculiar product feature able to enhance value addition is not exactly the consequence of a preliminary process choice, but the first reason, or one of the first reasons, for the definition of a peculiar food or non-food item. As a result, product diversification depends on commercial decisions based on consumeristic perceptions [10,13,28].

On these bases, a good portion of the above-mentioned products—coconut-related fluids only—will be analyzed in brief with reference to selected key points (the number and qualitative discussion of the processes will not be considered, with these features being defined as the effect of the below-mentioned factors):

(a) Packaging, durability, and size options;
(b) Sensorial features;
(c) Sustainability (eco-friendly products).

A summary of the qualitative evaluation is offered in Table 1.

**Table 1.** A qualitative evaluation of coconut-based beverages as value-added products.

| Original Coconut Raw Material | Value-Added Products | Packaging, Durability, and Size Options * | Sensorial Features * | Sustainability (Eco-Friendly Products) * |
|---|---|---|---|---|
| Coconut Water | Tender coconut water | | ++ | ++ |
| Coconut Water | Minimally processed tender coconut | | ++ | ++ |
| Coconut Water | Packaged tender coconut water | + | ++ | + |
| Coconut Water | Coconut water concentrates and frozen concentrates | + | + | + |
| Coconut Water | Bottled mature coconut water | + | + | + |
| Coconut Water | Coconut water beverages | | + | |
| Coconut WFL | Coconut WCF | ++ | | + |
| Coconut WFL | Coconut FWFL | + | + | + |
| Coconut WFL | Coconut PSM, syrups, and GPSM | + | + | + |
| Coconut Sap | Coconut jaggery, arrack | + | + | + |

* The "++" symbols mains "good or excellent result"; the "+" symbol" is for "intermediate or low/variable" results.

### 3.1. Packaging, Durability, and Size Options

Packaging options are discussed here, along with their relation to durability (low expectations: lower than three months; intermediate results: between three and 24 months; high performance: higher than 24 months)and size options (different available choices or one size/format only). From the viewpoint of value addition, and taking into account the problems of packaging, durability, and size choices at the same time, the best opportunities can be observed with reference to the following products, as shown in Table 1 [10,13,14].

(a) Packaged tender coconut water:Available packages are aluminum cans and coupled pouches. The choice of packages allows for the obtaining of technological results—in terms of reproducible, fast, and safety-acceptable production processes—and also for intermediate shelf-life expectation whencompared with only 24–36 h for fresh tender coconut water. Refrigerated conditions are a good option, but room temperature storage is also possible. Intermediate durability; different size options are available.

(b) Coconut water concentrates and frozen concentrates: Available packages are plastic containers and metal cans. Intermediate durability values range from 6 to 24 months on the basis of concentration. Different size options are available.

(c) Bottled mature coconut water: Glass or plastic bottles are available. Intermediate durability; different size options are available.

(d) Coconut water beverages: Glass or plastic bottles are available. Intermediate durability;.different size options are available.

(e) Coconut WCF: Sterilized tin cans are reported. Low, intermediate, and long durability; different size options are available.

(f) Coconut FWFL: Glass or plastic bottles are available. Low or intermediate durability; different size options are available.

(g) Coconut PSM, syrups, and GPSM: Glass bottles and plastic packages are available. Low or intermediate durability; different size options are available.

(h) Coconut jaggery, arrack:Glass bottles and plastic packages are available. Variable durability; different size options are available.

As a clear result, it appears that the greater part of the coconut-related beverages needs plastic coupled pouches, metallic cans, and glass containers. Shelf life expectations and the availability of different sizes are also correlated with the following points:

(1) Ameliorated aspect of the content (where possible; for example: transparent containers for caramelized/brown fluids).

(2) Enhanced shelf-life expectations (non-transparent packages are generally preferred with the aim of excluding ultraviolet rays and the consequent durability decay; metal cans are used at room temperature).

(3) Good sealability of packages.

(4) Good or excellent rapidity of packaging operations, especially where hot filling and other heat treatments are required, and high-speedprocessingis desirable.

*3.2. Sensorial Features*

From the viewpoint of value addition, and taking into account the problem of sensorial features (which should recall the original coconut nature), the following points can be highlighted, as displayed in Table 1 [10,13,14]:

(a)   Tender coconut water: Excellent features as ready-to-consume drink.
(b)   Packaged tender coconut water: Good or excellent sensorial performance if compared with tender coconut water.
(c)   Minimally processed tender coconut: Similar results in comparison with tender coconut water.
(d)   Coconut water concentrates and frozen concentrates: It is likely that sensorial features may depend on the subsequent use of these products; consequently, there is a risk of variable results. In any case, sensorial results should not be comparable to the original coconut water (moderate processing).
(e)   Bottled mature coconut water. Good or excellent results.
(f)   Coconut water beverages: Good or excellent results should be expected. However, sensorial results should be different from the original coconut water (with the addition of other compounds).
(g)   Coconut FWFL: Good results, also if used in conjunction with other products.
(h)   Coconut PSM, syrups, and GPSM. Good results, also if used in conjunction with other products.
(i)   Coconut jaggery, arrack: Good results are generally claimed.

As a clear result, it appears that the greater part of the value-added coconut-related beverages can exhibit good or excellent sensorial features when compared with the original raw materials. However, an interesting consideration concerns the "surrogate" nature of many of these products. In fact, the "coconut" origin is always claimed and recognized in the name of products; on the other hand, the trend appears to be towards the creation of articles which are able to replace, totally or partially, non-coconut-based foods. As a result, the sensorial evaluation does not appear to have a direct relationship with coconut, while similarities with other non-coconut-based foods are implicitly claimed (example: FWFL, alcoholic beverages, vinegar. . .). This factor should be taken into account when speaking of value addition.

*3.3. Sustainability (Eco-Friendly Products)*

From the viewpoint of value addition, and taking into account the problem of sustainability and eco-friendly products and services, the basic definition of the Sustainable Development Goals (SDG) as defined at the United Nations Conference on Sustainable Development Rio+20 (Rio de Janeiro, Brazil, 2012) should be remembered. In this ambit, 17 different SDG were defined, including [33] the following:

(a)   "End hunger, achieve food security and improved nutrition, and promote sustainable agriculture" (SDG No 2).
(b)   "Ensure availability and sustainable management of water and sanitation for all" (SDG No 6).
(c)   "Ensure sustainable consumption and production patterns" (SDG No 12).

On these bases, and taking into account that the supply and productive chain of coconut-based products is still rather low when compared with other food production ambits, the following aspects can be highlighted (Table 1) when speaking of sustainable/eco-friendly coconut-related products [10,13,14]:

(a)   The only sustainable and eco-friendly beverages are naturally tender coconut water and minimally processed tender coconut water (with absent or very limited food processing; the absence of packaging materials).
(b)   All processed coconut-based beverages suffer of the same limitation of other industrial foods. It is likely that enhanced durability will be a good challenge because intermediate- and long-durability foods are correlated with moderate or strong pro-

cessing degrees, enhanced packaging materials, high-energy consuming storage systems, and broad transportation networks. The use of glass bottles and recyclable containers can be a distinctive advantage in this ambit, while plastic-based packaging materials and objects are still a recovery/recycling problem. Consequently, all remaining value-added products may have interesting margins for improvement from a sustainability viewpoint and, consequently, an intermediate impact.

An important opportunity could be the option for organic and fair-trade coconut productions. At present, there is some difficulty in these ambits [34–38]. However, value addition should receive a notable enhancement in the specific sector of coconut-based beverages in the future. Also, because there is some parallelism between these policies (the Roundtable on Sustainable Palm Oil, www.rspo.org, accessed on 10 October 2023, has had reputable success) and sustainable models, more efforts are needed.

### 4. Coconut-Based Beverages and Value-Added Products: Concluding Remarks

The aim of this paper has been to show the current perspectives on value addition with reference to coconut-based beverages. After a detailed discussion concerning all typologies of coconut-derived products, including solid commodities, consideration of the basic value addition drivers for beverages has shown that qualitative differences between these products are not strictly dependent on the extent of the processing degree. In fact, the higher the expectation of designers, industries, and finally consumers when speaking of durable, processable, and palatable foods, the higher the number of different processing and preservation treatments. Consequently, the processing degree is a consequence of the consumeristic request. As a result, our investigation has mainly taken into account three different value addition drivers which can be directly managed by coconut harvesters, producers, and so on: packaging, durability, and size options; sensorial features; and sustainability (eco-friendly products). The results of this investigation have highlighted the notable added value of several products—packaged tender coconut water; coconut water concentrates and frozen concentrates; bottled mature coconut water; coconut water beverages; coconut WCF; coconut FWFL; coconut PSM, syrups, and GPSM; coconut jaggery, arrack—because of their recyclable packaging materials (even if recovery and recycling are still a big challenge when speaking of plastic packages), intermediate- or long-durability expectations, different available sizes, and good or excellent sensorial performances. Sustainability and eco-friendly policies may be a problem for those products which are produced similarly to non-coconut-based beverages (and the trend in favor of extensive research for surrogates in the food industry has been noted), but the opportunity of certified organic and/or fair-trade products could help the coconuts industry in the near future, similarly to successes in similar ambits. In any case, more research is still needed when speaking of value addition for coconut-based beverages.

**Author Contributions:** Conceptualization, S.P. and S.M.V.; methodology, S.P. and C.P.; analysis, S.P. and C.P.; investigation, S.P. and C.P.; resources, S.P. and C.P.; data curation, S.P. and S.M.V.; writing—original draft preparation, S.P. and C.P.; writing—review and editing, S.P. and S.M.V.; supervision, S.P.; project administration, S.P. and S.M.V. All authors have read and agreed to the published version of the manuscript.

**Funding:** This research received no external funding.

**Data Availability Statement:** No new data were created or analyzed in this study. Data sharing is not applicable to this article.

**Conflicts of Interest:** The authors declare no conflicts of interest.

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
