# Peer review of "Value Addition and Coconut-Based Beverages: Current Perspectives"

_beverages, doi:10.3390/beverages10010014_

Round 1

Reviewer 1 Report

Comments and Suggestions for Authors

The review paper entitled “Value Addition and Coconut-based Beverages. Current Perspectives”  discusses the perspectives of value addition in the context of coconut-based beverages. It covers various aspects such as the definition of value addition, the challenges and opportunities in the food and beverage industry, and specific considerations for coconut-derived products. The discussion includes factors like processing degree, diversification, marketing opportunities, and the role of different product features such as packaging, durability, size options, sensorial features, and sustainability. It emphasizes the importance of product diversification and how it influences economic gains. Additionally, the discussion touches upon the challenges and opportunities related to sustainability and eco-friendly practices in the coconut industry.

The content of the paper is significant and I believe it should be accepted for publications after minor revision. 

1. The Table is referred as table 1 in the text but in the legent is named Table 2. It should be corrected. I also suggest that it should be moved in the paragraph of the text where it is mentioned for the first time, to be more readers friendly. 

2. Authors must check some grammar mistakes especially in paragraph 4. Coconut-based Beverages and Value-added Products. Concluding Remarks like de-signers, (line 446)

Author Response

Dear Reviewer, thank You very much for valuable recommendations. We have:

  1. Renamed Table 2 (as Table 1) and placed in the text immediately after Section 3 (before Sect. 3.1);
  2. Checked grammar mistakes, as required.

You can find all variations in the new amended file. Please see the attachment.

Kind Regards,

The Authors

Reviewer 2 Report

Comments and Suggestions for Authors

Dear authors 

Thank you so much for a well-structured manuscript. This is an interesting and trendy topic. What I have found from this manuscripts are: 

- 50% of the manuscript is the definition of coconut beverage and processed coconut-based beverage.

Three different drivers for value addition haven't contained enough strong support or references. There was no data support on this claims/ conclusion.

- It will be more useful if the difference of value-added coconut-based beverages is discussed in the chemical composition, the impact of process on beverage quality, the health benefits, marketing perspectives etc. 

I hope my comments are helpful for the team to improve the manuscript. 

Author Response

Dear Reviewer, thank You very much for valuable recommendations and comments. Here You are our answers:

  1. We are aware that 50% of the manuscript describes and discusses definitions of coconut beverage and processed coconut-based beverage. On the other hand, the aim of this work is to define some possible pathways for the analysis and future improvements concerning these products both on the technological ground and the economic ground. Being the text a partial analysis of economic trends and value additions, we suppose the definition of these products has to be clearly discussed and given to a vast audience, including non-expert Readers in technological or economically relevant ambits. Consequently, our opinion is that a notable portion of the text is needed in this form;
  2. The discussion concerning drivers for value additions is a research work based on different research papers (10, 13 to 14, 35 to 39). Actually, some of these research papers have already published a notable amount of data which can be summarized in form of three different drivers, depending on technological aspects and economic evaluations. In addition, the mere repetition and citation of already cited reference is not practicable. As an example, the following reference:

Naik, A.; Madhusudhan, M. C.; Raghavarao, K. S. M. S.; Subba, D. Downstream processing for production of value added products from coconut. Curr. Biochem. Eng.2015, 2, 168-180

reports different data and concepts, including economic reasons for value-added products, from 71 additional papers.

On these bases, we suppose the discussion is supported enough, while the existence of further drivers has to be investigated at present, suggesting more research is needed;

3. The difference of value-added coconut-based beverages can be discussed on the basis of quantitative and qualitative parameters. On the one side, there are the chemical composition, the impact of processes on beverage quality, health and nutritional benefits, etc. These factors are strictly correlated and referred or able to determine packaging strategies, durability options, product weights, product brands, sensorial features, etc. In other terms, the last points are function and result (effects) of the first technological drivers. Consequently, there is not necessity of defining the first parameters point by point with the aim of “translating” them in terms of final effects. As an example, chemical composition influences packaging, durability, weights, and sensorial/nutritional features; consequently, differences can be discussed in terms of the final effect(s), also because cited references have surely explained basic chemical features in a better way than in a repeated analysis. Anyway, each product category is also explained with some reference to these topics, with reference to the main features which could justify some interest and ameliorated performance against non-coconut based beverages.

On the other side,  there are “other factors not directly related to products, production processes, and so on, including the geographical position of markets, regulatory restrictions, advertisement strategies, consumeristic perception of quality/price ratios (in terms of supposed quality performances), beliefs, religions, different norms, etc.” These points can be discussed in terms of marketing perspectives, similarly to some of cited references. Anyway, “the aim of this paper is to examine the situation of coconut-based beverages, including also different products which can be able to detract the amount of marketable beverages in favor of other market channels (and related consumers)”.  In this ambit,  it should be clear enough that “an interesting consideration concerns the ’surrogate’ nature of many of these products. In fact, the ‘coconut’ origin is always claimed and recognized in the name of products; on the other side, the trend appears to be the creation of articles which can be able to replace totally or partially non-coconut based foods. As a result, the sensorial evaluation does not appear to have direct relationship with coconut, while the similarity with other non-coconut based foods is implicitly claimed (…). This factor should be taken into account when speaking of value addition”.

In other words, the value addition problem for coconut-based beverages is not strictly linked with chemical composition, health and nutritional evaluation, etc. because all of these products are mainly produced with the aim of acting as surrogates for non-coconut based products. Substantially, coconut-based beverages can prevail on the market (against non-coconut-based beverages) if additional drivers are used. Nutritional, health, and chemical profiles of these products are not strictly better than non-coconut based beverages (and these profiles have briefly cited for each category). For these reasons, additional drivers have to be used.

As a result, our opinion is that the review has explained the situation in a comprehensible way, also taking into account briefly essential chemical, health, nutritional, and marketing considerations and profiles.

Kind Regards,

The Authors

Reviewer 3 Report

Comments and Suggestions for Authors

Interesting work, it fits into the current trend of innovative food. However, I have a very serious concern about the nomenclature used for coconut-based products. The author describes flours, yogurts, jams, margarines, etc. made from coconut, however, the author must also take into account EU legislation in the article, where the mentioned products have specific definitions and e.g. milk obtained from coconut cannot be called coconut milk. In the reviewer's opinion, this must be included because it misleads readers. 

Author Response

Dear Reviewer,

Thank You very much for valuable recommendations and comments. Here You are our answer. The concern has a reliable basis with reference to current (and future) regulatory norms, in the EU and in other Nations (India, Australia, etc.). On the one side, pursuant to Regulation (EU) No 1308/2013, the terms ‘milk’, ‘cheese’, ‘butter’, ‘cream’ are reserved for diary product (‘dairy’ names).  The question whether ‘diary’ names could be used for diary alternatives has been discussed by the Court of Justice of European Union in 2017: the Court ruled that the reserved dairy names cannot be used even if combined with clarifying explanations as they cannot prevent confusion in the mind of the consumer with certainty.

On the other side, in Decision 2010/791/EU, European Commission granted exceptions for products, the nature of which is clear from the traditional use. Such products are peanut butter, coconut milk, creamed coconuts, etc. As an example, “coconut milk” is accepted in the U.S… So, the reference to certain products previously named  in scientific papers as milk, creams, yoghurts, and other products, may be difficult enough.

Based on this situation, we propose to modify the names of products by avoiding non-dairy and dairy-related names with potential misleading meanings,  and with a clear relation to original references where these products are differently named.  The modification of names is justified in the text with a short clarification concerning non-dairy products in the EU.

The following modifications have been applied:

Coconut milk becomes “coconut white fatty liquid” (coconut WFL)

Coconut cream becomes “coconut white condensed fat” (coconut WCF)

Coconut jam becomes “coconut pulp and sugars mixture” (coconut PSM)

Coconut honey becomes “golden coconut pulp and sugars mixture” (coconut GPSM)

Coconut yoghurt becomes “fermented coconut white fatty liquid” (coconut FWFL)

Coconut margarine becomes “coconut unsaturated fat-oil blend” (coconut UFOB)

Coconut flour becomes “coconut dried and ground pulp” (coconut DGP).

You can find all amendments in the new amended file.

Kind Regards,

The Authors
